# One-dimensional carbon chains encapsulated in hollandite

Jonathan M. Polfus [1✉]

One-dimensional carbon chains are highly reactive allotropes that are stabilized inside the protective environment of carbon nanotubes. Here we show that carbon chains can be encapsulated in metal oxides containing open structural channels, exemplified by hollandite $\alpha$-$MnO_2$. The $\alpha$-$MnO_2$ channels stabilize cumulene chains due to their structural commensurability, whereas the triple bonds in polyyne chains exhibit excessive steric repulsion to the oxide ions bordering the channel. Cumulene exhibits an interaction energy of only 0.065 eV per carbon atom, obtained by first-principles calculations, which is significantly more favorable than for encapsulation in a similarly sized carbon nanotube. Encapsulation of carbon chains is associated with lateral expansion of the $\alpha$-$MnO_2$ channel and polarization of the manganese and oxygen charge densities adjacent to the chains. Accordingly, the interaction energy is governed by a balance between van der Waals attraction and steric repulsion between the materials.

[1] Department of Chemistry, Centre for Materials Science and Nanotechnology, University of Oslo, PO Box 1033 Blindern N-0315 Oslo, Norway.
✉email: jonathan.polfus@kjemi.uio.no

The rich chemistry of carbon gives rise to a range of allotropes that can be characterized according to the hybridization of the atomic orbitals and corresponding spatial arrangement of chemical bonds: three-dimensional (3D) $sp^3$ hybridization in diamond; two-dimensional (2D) $sp^2$ hybridization in graphite and its derivatives, graphene, carbon nanotubes and fullerenes. Finally, one-dimensional (1D) sp hybridization results in linear carbon chains referred to as carbyne or cyclic carbon chains. As an allotrope and material analogous to diamond and graphite, carbyne can be defined as a van der Waals crystal comprising a hexagonal lattice of carbon chains[1].

Linear carbon chains can exist in two forms. Cumulene has consecutive double bonds, i.e., $(=C=C=)_n$, and exhibits metallic character due to two degenerate π bands that are half-occupied. This makes cumulene susceptible to Peierls distortion[2] and conversion to the polyyne form with alternating single and triple bonds, i.e., $(-C \equiv C-)_n$, and corresponding bond length alteration (BLA)[3]. The polyyne form exhibits a band gap and represents the ground state structure[4,5]. However, due to its chemical activity and extreme instability in ambient conditions, indications of naturally formed carbyne have only been observed in interstellar dust and meteorites[6–8]. Carbyne was first synthetically realized inside the protective environment of multiwalled carbon nanotubes[9]. Shi et al. established a synthesis route for preparing carbyne chains of more than 6000 atoms inside double-walled carbon nanotubes[10].

First-principles calculations have revealed favorable interaction energies for the encapsulation of carbyne into carbon nanotubes, e.g., approx. −0.2 eV per carbon atom encapsulated in (penta-graphene) nanotubes with diameters of approx. 7–8 Å[10–12]. These interactions can originate from van der Waals forces as well as charge transfer. While the van der Waals contributions are not necessarily accounted for in first-principles calculations since they require explicit treatment, charge transfer has been reported to occur from carbon nanotubes to carbyne, e.g., approx. 0.02–0.05 electrons per carbon in polyyne chains[13,14]. Consequently, the bond length alteration and phonon frequencies of carbyne are affected by encapsulation in nanotubes or adhesion to graphene layers[14–16].

In the solid phase, carbyne has been studied as a fiber material in metal matrix nanocomposites[17], in addition to being the constituent of van der Waals crystals[1]. Here, we investigate the prospect of a new class of hybrid nanocomposite materials comprising carbyne encapsulated in a metal oxide matrix. To this end, hollandite structured α-MnO$_2$ represents an intriguing candidate material due to its open channels with a diameter of approx. 4.9 Å between diagonally protruding oxide ions (Fig. 1a). These channels enable intercalation of relatively large alkali cations ($Na^+$, $K^+$, $Ba^{2+}$) that become coordinated by lattice oxygen and intercalated water, if present[18].

Density functional theory (DFT) simulations were performed using the state-of-the-art SCAN+rVV10 van der Waals functional[19–21], which simultaneously provides a reliable description of the structure, electronic properties and thermodynamic stability of MnO$_2$ polymorphs[22] and accounts for dispersion interactions between carbyne and the encapsulating oxide material[23]. Cumulene was modeled as a periodic C$_9$ chain (Fig. 1b). Notably, the supercells were constructed with minor lattice mismatches, i.e., −1.28% for cumulene and 0.13% for α-MnO$_2$ in the composite supercell relative to the pristine materials (Supplementary Note 1). The polyyne form of carbyne can be modeled as acetylenic molecules, i.e., polyyne chains terminated by hydrogen or other groups; C$_6$H$_2$ was used here (Fig. 1c).

The electronic structure simulations reveal that carbyne can be encapsulated in the channels of α-MnO$_2$ with an interaction energy that is more favorable than for encapsulation in a carbon nanotube with a slightly larger diameter. Moreover, α-MnO$_2$ provides a chemical environment that stabilizes the otherwise elusive cumulene form of carbyne.

## Results and discussion

The triple bonds in C$_6$H$_2$ exhibit spatially expanded charge densities that are sterically repulsed by the oxide ions bordering the α-MnO$_2$ channels. Thus, the single bonds in C$_6$H$_2$ coincide with the protruding oxide ions in the optimized structure, as shown in Fig. 2a. This alignment is, however, not possible for longer polyynic chains due to the different periodicity of the two structures. The extent of the steric repulsion between the triple bonds and the protruding oxide ions can be probed by assessing the energy barrier for the displacement of C$_6$H$_2$ along the channel. As shown in Fig. 3, this energy barrier is quite large and amounts to about 1 eV. In the optimized configuration, C$_6$H$_2$ exhibits a slightly unfavorable interaction energy of 0.090 eV per carbon atom relative to the isolated materials (Table 1).

In comparison to the triple bonds of C$_6$H$_2$, the charge density of cumulene is essentially homogenous along the chain (Fig. 2b). The calculated energy barrier for the displacement of cumulene

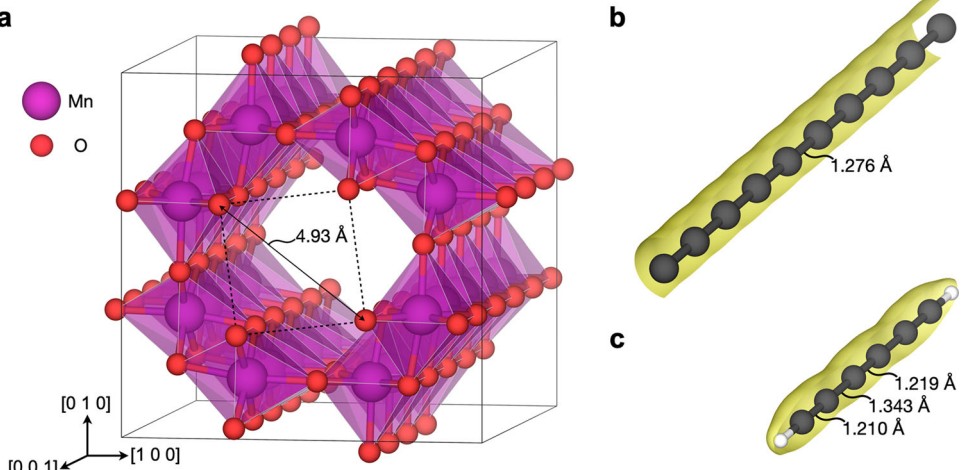

**Fig. 1 Structure of α-MnO$_2$ and carbon chains. a** Optimized structure of α-MnO$_2$ (1 × 1 × 4 supercell). The dotted lines indicate the protruding oxide ions that represent the narrowest regions of the channel. **b** Cumulene modeled as a periodic C$_9$ chain. **c** Polyyne modeled as a C$_6$H$_2$ molecule. Charge density isosurfaces are shown in yellow.

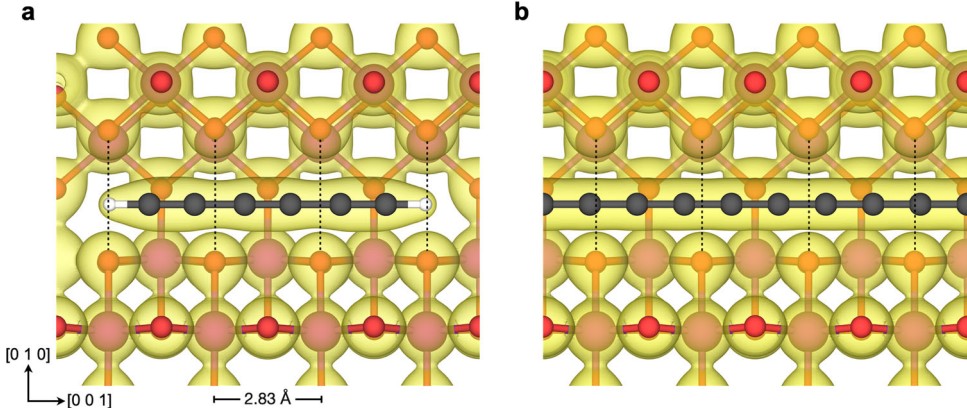

**Fig. 2 Cross-section views of α-MnO₂ with encapsulated chains. a** Polyyne $C_6H_2$ and **b** cumulene $C_9$, with charge density isosurfaces ($0.1\ a_0^{-3}$). The dotted lines indicate the protruding oxide ions that represent the narrowest regions of the channel.

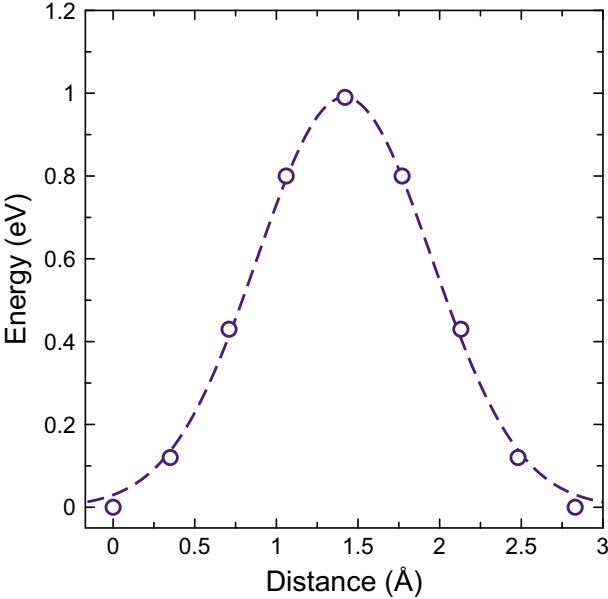

**Fig. 3 Energy barrier for displacement of $C_6H_2$ along the c-axis of α-MnO₂.** The displacement distance of 2.83 Å corresponds to the periodicity of the channels (Fig. 2a). The dashed line is a Gaussian fit.

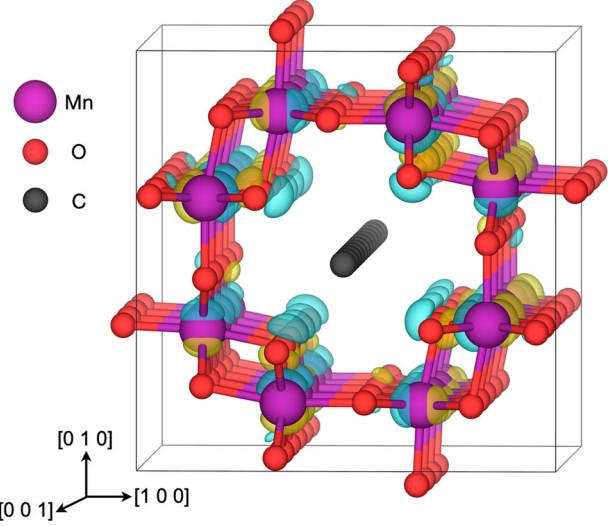

**Fig. 4 Charge density difference upon encapsulation of cumulenic $C_9$ in α-MnO₂.** Blue and yellow isosurfaces ($0.05\ a_0^{-3}$) represent negative and positive charge density differences, respectively.

**Table 1 Interaction energy and structural changes upon encapsulation in α-MnO₂ and carbon nanotubes.**

| Material | Diameter (Å) | Lateral expansion (%) | Interaction energy (eV) |
|---|---|---|---|
| α-MnO₂ + $C_6H_2$ | 5.04 | 0.46 | 0.090 |
| α-MnO₂ + $C_9$ | 5.08 | 0.92 | 0.065 |
| CNT (7,1) + $C_{17}$ | 5.99 | 0.00 | 0.198 |
| CNT (5,5) + $C_{19}$ | 6.84 | 0.00 | −0.194 |

The interaction energies are given per carbon atom along with the expansion of the lateral supercell parameters and diameter of the encapsulating channel.

along the c-axis of α-MnO₂ is therefore miniscule (1.2 meV, Supplementary Fig. 1). Notably, cumulene exhibits an interaction energy to the encapsulating oxide of only 0.065 eV per carbon atom (Table 1). Steric interactions between cumulene and α-MnO₂ are alleviated by the expansion of the lateral supercell parameters by 0.92% and an increase in the channel diameter to 5.08 Å. The lower interaction energy and better commensurability

between the structures imply that α-MnO₂ stabilizes the cumulene form of carbyne.

The interaction energy for cumulene is significantly more favorable than for encapsulation in a carbon nanotube (CNT) with chiral indices (7,1) and a slightly larger diameter of 5.99 Å (Table 1). The interaction energy between cumulene and the considerably larger (5,5) nanotube is approx. −0.2 eV per carbon atom in line with previous studies[10–12]. In contrast to α-MnO₂, the structural rigidity of the nanotubes inhibits their lateral expansion upon encapsulation of carbon chains.

Chemical interactions between the carbon chains and the metal oxide were evaluated based on changes in the charge densities upon encapsulation. The absence of charge density differences associated with the carbon chains indicates that there is no significant transfer of charge between the materials, in accordance with Bader charge analysis. However, the charge densities of the protruding oxide ions were reduced in the direction of the carbon chains and the adjacent manganese cations showed a corresponding polarization, as shown for encapsulated cumulene in Fig. 4. Encapsulated $C_6H_2$ showed similar polarizations of the charge density although it was confined to the regions with the largest steric repulsion, i.e., adjacent to the triple bonds

(Supplementary Fig. 2). Moreover, changes in zero-point vibrations of the encapsulating oxide were minor, e.g., the zero-point energy of a protruding oxide ion was reduced by 0.7 meV upon insertion of cumulene (Supplementary Table 3).

## Conclusions

Metal oxides with structural channels can accommodate carbon chains with relatively low interaction energies, as exemplified by the encapsulation of cumulenic $C_9$ in α-$MnO_2$. The interaction energy of 0.065 eV per carbon atom is more favorable than for encapsulation in the (7,1) carbon nanotube despite the smaller channel diameter in α-$MnO_2$. The chemical environment of the α-$MnO_2$ channels stabilizes the cumulene form of carbyne due to better commensurability between the structures as well as the more favorable interaction energy compared to polyenic $C_6H_2$. Encapsulation of carbon chains is associated with lateral expansion of the α-$MnO_2$ channel, polarization of the charge densities of oxygen closest to the chain, and otherwise minor changes with respect to charge transfer between the materials.

## Methods

DFT calculations were carried out using VASP 6.3 with projector-augmented wave pseudopotentials and the SCAN+rVV10 van der Waals functional[19–21]. The calculations were spin-polarized with an explicit treatment of the following valence electrons: Mn $3s^2p^6d^5 4s^2$; O $2s^2p^4$; C $2s^2p^2$ and H $1s^1$. The plane wave energy cutoff was 500 eV, and the k-point grid was $4 \times 4 \times 16$ grid for the cubic α-$MnO_2$ unit cell. The optimized lattice parameters of α-$MnO_2$ were a = 9.637 Å and c = 2.835 Å. Antiferromagnetic ordering was imposed with ferromagnetic coupling between two neighboring c-axis columns of edge-sharing manganese octahedra and alternation of these double columns[24,25].

Cumulene or polyenic $C_6H_2$ was introduced into one out of two channels in $1 \times 1 \times 4$ supercells of α-$MnO_2$ (96 atoms). The isolated polyyne $C_6H_2$ molecule was modeled in vacuum using cubic cells of 20 Å and Γ-point sampling. Periodic $C_9$ was modeled in vacuum with lateral cell parameters of 20 Å and a $1 \times 1 \times 4$ k-point grid. Interaction energies were calculated as the total energy difference between the composite supercell and the isolated materials and normalized per carbon atom. The lattice parameters of the composite cell were imposed on the isolated materials in order for the interaction energy to not include the strain induced in both materials due to the periodic boundary conditions. The activation energy for the diffusion of carbyne along the α-$MnO_2$ channels was investigated by static displacement along the channel and the climbing image nudged elastic band (CI-NEB) method[26]. Additional details are provided in Supplementary Methods.

## Data availability

The data generated in this study are available in the figshare repository, https://doi.org/10.6084/m9.figshare.24132285.v1.

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

## Acknowledgements

The author acknowledges the Research Council of Norway for financial support (Grant No. 262274) and Uninett Sigma2 for computational resources (NN4604K). Dr. Wen Xing (SINTEF, Norway) is thanked for the fruitful discussions.

## Author contributions

J.M.P.: conception, funding, design, simulations, data analysis and writing of the manuscript.

## Competing interests

The author declares no competing interests.
