## [Peer Review File · Communications Chemistry]

Reviewers' comments:

Reviewer #1 (Remarks to the Author):

In this manuscript, the author investigated encapsulation of linear carbon molecules (polyynes C₆H₂ and cumulene C₉) inside the linear hole channels of α -MnO₂ crystal by first-principles DFT calculations. As a result of the total energy comparison, the author concluded that cumulene can be encapsulated in α -MnO₂ but C₆H₂ cannot. This is a very interesting result, if it is correct. However, I have some doubt about the AFM configuration that the author assumed. In Method section, the author wrote that antiferromagnetic ordering was retained with alternating ferromagnetic columns along the c-axis in all optimized calculations, by referring to Refs. [17] and [21] of the previous first-principles calculations of the α -MnO₂ phase. Although the author seems to be not aware of another paper, Phys. Chem. Chem. Phys. 18, 13294 (2016). DOI: 10.1039/C5CP07806G, the relative total energies of several AFM configurations of the α -MnO₂ phase were compared in Table S1 in supplementary information (SI) of this paper. The AFM configuration written in Refs. [17] and [21] corresponds to AFM4 (Fig. S1(c)) of this SI, and is 0.007 eV higher in energy than the most stable AFM (Fig. 2(b)) of this paper according to this SI. Therefore, I suspect, there is a possibility that the AFM configuration that the author assumed is not the most stable one. In this respect, I am wondering if the author's result might change according to the AFM configurations of the α -MnO₂ phase. So, before this manuscript is considered for publication, I want to request the author to check the AFM configurations of the α -MnO₂ phase and if the result changes or not according to the AFM configurations.

Reviewer #2 (Remarks to the Author):

This work reports a theoretical investigation of the carbon chains encapsulated in α -MnO₂ using DFT calculations. The author shows that the channel of α -MnO₂ can accommodate carbon chains, and the interaction energy between α -MnO₂ and carbon chains is determined by the competing effects of the van der Waals attraction and steric repulsion. There are several points that require further clarification.

1. Does polar MnO₂ channel have distinct behaviors different from CNTs? It would be interesting to compare these two polar and nonpolar examples theoretically from chemical bonding of view.
2. Compared with carbon chains encapsulated in CNTs, the hollow core of α -MnO₂ (~5 Å) is even smaller. According to Table S1, the cell parameters along the a-axis are enlarged. Does this enlargement indicate a strong repulsive interaction between C₉ and α -MnO₂ due to van der Waals forces? How much does this interaction affect the C-C bond length of C₉ compared with isolated cumulene as shown in Figure 1b? Can the SCAN+rVV10 functional accurately capture the changes in the C-C bond length of C₉?
3. There are slight lattice mismatches between α -MnO₂ and C₉, and the calculated binding energy is also small (-0.05 eV/C). To what extent does the lattice mismatch impact the energy of the composite system, and does it affect the qualitative results?

4. How does the axial position of C9 affect the energy of the composite system? Does it exhibit energy barriers for displacement similar to the ones observed in C6H2, as shown in Figure 3?

5. Table S2 indicates relatively large changes in vibrational frequencies along the c-axis, suggesting that encapsulation affects the zero-point vibrations of oxygen anions. Did the author also calculate the zero-point vibrations of the C9 chain, and if so, how does it impact Raman activity?

6. In addition to first-principles calculations, MD results have also confirmed that carbon chains can be inserted into CNTs with diameters favorable for vdW interactions. For example, relevant publications such as "Nanoscale, 2023, 15, 6143 (10.1039/D3NR00386H)" and "Carbon 2021, 183, 571 (10.1016/j.carbon.2021.07.037)" provide further support for this observation and could be referenced in the introduction section.

Response to the Reviewers

The author is grateful for the reviewers' comments and suggestions that have helped to improve the scientific quality and clarity of the manuscript. The comments have been addressed by performing additional simulations, and the results have been added in the form of figures, tables and changes to the manuscript files.

The reviewers' comments and point-by-point responses are included below with the related changes made to the manuscript and supplementary information. Other minor changes are highlighted in the manuscript file.

Reviewer #1

In this manuscript, the author investigated encapsulation of linear carbon molecules (polyene C_6H_2 and cumulene C_9) inside the linear hole channels of α - MnO_2 crystal by first-principles DFT calculations. As a result of the total energy comparison, the author concluded that cumulene can be encapsulated in α - MnO_2 but C_6H_2 cannot. This is a very interesting result, if it is correct. However, I have some doubt about the AFM configuration that the author assumed. In Method section, the author wrote that antiferromagnetic ordering was retained with alternating ferromagnetic columns along the c-axis in all optimized calculations, by referring to Refs. [17] and [21] of the previous first-principles calculations of the α - MnO_2 phase. Although the author seems to be not aware of another paper, Phys. Chem. Chem. Phys. 18, 13294 (2016). DOI: 10.1039/C5CP07806G, the relative total energies of several AFM configurations of the α - MnO_2 phase were compared in Table S1 in supplementary information (SI) of this paper. The AFM configuration written in Refs. [17] and [21] corresponds to AFM4 (Fig. S1(c)) of this SI, and is 0.007 eV higher in energy than the most stable AFM (Fig. 2(b)) of this paper according to this SI. Therefore, I suspect, there is a possibility that the AFM configuration that the author assumed is not the most stable one. In this respect, I am wondering if the author's result might change according to the AFM configurations of the α - MnO_2 phase. So, before this manuscript is considered for publication, I want to request the author to check the AFM configurations of the α - MnO_2 phase and if the result changes or not according to the AFM configurations.

Author reply: Thank you for identifying this important issue. As suggested, additional simulations have been performed with this most stable AFM configuration, and fortunately, the results are very similar for the interaction energy between C_9 and α - MnO_2 . At the same time, a minor error in the reference energy for C_9 was identified which resulted in a slight shift of the interaction energies. Nevertheless, the main conclusions remain the same, in particular when taking into account the direct comparisons with encapsulation in carbon nanotube (see Referee 2, question 1).

The interaction energies and corresponding structural features have been updated with the new AFM configuration in the main manuscript. The comparisons between the AFM configuration are included in the supplementary information:

“The interaction energy between cumulenic C₉ and the encapsulating oxide was similar for both types of antiferromagnetic ordering, i.e., 0.065 eV and 0.056 eV and for AFM1 and AFM2, respectively.”

The description of the AFM configuration (and corresponding references) has been included in the methods section of the main manuscript:

“Antiferromagnetic ordering was imposed with ferromagnetic coupling between two neighboring c-axis columns of edge-sharing manganese octahedra and alternation of these double columns.^{1,2}”

- 1 Crespo, Y. & Seriani, N. Electronic and magnetic properties of α -MnO₂ from *ab initio* calculations. *Physical Review B* **88** (2013). <https://doi.org:10.1103/PhysRevB.88.144428>
- 2 Noda, Y., Ohno, K. & Nakamura, S. Momentum-dependent band spin splitting in semiconducting MnO₂: a density functional calculation. *Phys Chem Chem Phys* **18**, 13294-13303 (2016). <https://doi.org:10.1039/c5cp07806g>

The methods section in the supplementary information includes a definition of the AFM configurations and the energy difference between these were determined to be exactly as reported in the suggested reference:

“With the current computational parameters, the latter configuration was more stable by -0.0007 eV per manganese atom, in complete agreement with the results of Noda et al.²”

Reviewer #2

This work reports a theoretical investigation of the carbon chains encapsulated in α -MnO₂ using DFT calculations. The author shows that the channel of α -MnO₂ can accommodate carbon chains, and the interaction energy between α -MnO₂ and carbon chains is determined by the competing effects of the van der Waals attraction and steric repulsion. There are several points that require further clarification.

1. Does polar MnO₂ channel have distinct behaviors different from CNTs? It would be interesting to compare these two polar and nonpolar examples theoretically from chemical bonding of view.

Author reply: In order to compare these systems, additional simulations have been performed on encapsulation of cumulene in two different carbon nanotubes with diameters comparable or slightly larger than the α -MnO₂ channels. These interaction energies have been included in Table 1 in the main manuscript and compared to encapsulation in α -MnO₂:

“The interaction energy for cumulene is significantly more favorable than for encapsulation in a carbon nanotube (CNT) with chiral indices (7,1) and a slightly larger diameter of 5.99 Å (Table S2). The interaction energy between cumulene and the considerably larger (5,5) nanotube is approx. -0.2 eV per carbon atom in line with previous studies.³⁻⁵ In contrast to α -MnO₂, the structural rigidity of the nanotubes inhibit their lateral expansion upon encapsulation of carbon chains.”

Table 1: Interaction energy and structural changes upon encapsulation in α -MnO₂ and carbon nanotubes. The interaction energies are given per carbon atom along with the expansion of the lateral supercell parameters and diameter of the encapsulating channel.

Material	Diameter (Å)	Lateral expansion (%)	Interaction energy (eV)
α -MnO ₂ + C ₆ H ₂	5.04	0.46	0.090
α -MnO ₂ + C ₉	5.08	0.92	0.065
CNT (7,1) + C ₁₇	5.99	0.00	0.198
CNT (5,5) + C ₁₉	6.84	0.00	-0.194

- 3 Kuwahara, R., Kudo, Y., Morisato, T. & Ohno, K. Encapsulation of carbon chain molecules in single-walled carbon nanotubes. *Journal of Physical Chemistry A* **115**, 5147-5156 (2011). <https://doi.org:10.1021/jp109308w>
- 4 Rocha, R. A., Santos, R. B. D., Junior, L. A. R. & Aguiar, A. L. On the Stabilization of Carbynes Encapsulated in Penta-Graphene Nanotubes: a DFT Study. *J Mol Model* **27**, 318 (2021). <https://doi.org:10.1007/s00894-021-04918-7>
- 5 Shi, L. *et al.* Confined linear carbon chains as a route to bulk carbyne. *Nat Mater* **15**, 634-639 (2016). <https://doi.org:10.1038/nmat4617>

The CNT cells were constructed with minor strain, as described in the in the Methods section of the supporting information:

“Carbon nanotube with chiral indices (7,1) and (5,5) were modelled in vacuum with lateral cell parameters of 20 Å and a lengthwise expansion by 2 and 10 unit cells, respectively. The k-point grids were 1 × 1 × 2. The optimized lattice parameters were 20.06 Å for the (7,1) nanotube

and 24.56 Å for the (5,5) nanotube, yielding strains of -1.09% and 1.25% for encapsulated C₁₇ and C₁₉ cumulene chains, respectively. “

2. Compared with carbon chains encapsulated in CNTs, the hollow core of α -MnO₂ (~5 Å) is even smaller. According to Table S1, the cell parameters along the a-axis are enlarged. Does this enlargement indicate a strong repulsive interaction between C₉ and α -MnO₂ due to van der Waals forces? How much does this interaction affect the C-C bond length of C₉ compared with isolated cumulene as shown in Figure 1b? Can the SCAN+rVV10 functional accurately capture the changes in the C-C bond length of C₉?

Author reply: The lateral expansion and relaxed channel diameter of α -MnO₂ upon encapsulation of carbon chains has been included in Table 1 in the main manuscript (see previous comment):

“Steric interactions between cumulene and α -MnO₂ are alleviated by expansion of the lateral supercell parameters by 0.92% and an increase in the channel diameter to 5.08 Å. The lower interaction energy and better commensurability between the structures implies that α -MnO₂ stabilizes the cumulene form of carbyne.”

The C₉ chain becomes compressively strained upon encapsulation by -1.28% as noted in the introduction. The corresponding change in the C-C bond length has been added to the supplementary information:

“Notably, a periodic chain of nine cumulenic carbon atoms shows a minor compressive strain of -1.28% with the relaxed c-axis of the composite supercell (**Error! Reference source not found.**), corresponding to a change in the C-C bond length from 1.276 Å to 1.262 Å.”

Finally, the following reference has been added as Ref. 23 to attest to the performance of the SCAN functional in describing diversely bonded systems, including C-C bonds.

J. Sun, R. C. Remsing, Y. Zhang, Z. Sun, A. Ruzsinszky, H. Peng, Z. Yang, A. Paul, U. Waghmare, X. Wu, M. L. Klein and J. P. Perdew, *Nat Chem*, 2016, 8, 831-836.

3. There are slight lattice mismatches between α -MnO₂ and C₉, and the calculated binding energy is also small (-0.05 eV/C). To what extent does the lattice mismatch impact the energy of the composite system, and does it affect the qualitative results?

Author reply: Effects of lattice mismatch are removed from the calculation of the interaction energy, and this has been clarified in the Methods section:

“Interaction energies were calculated as the total energy difference between the composite supercell and the isolated materials. The lattice parameters of the composite cell were imposed on the isolated materials in order for the interaction energy to not include the strain induced in both materials due the periodic boundary conditions of the cells.”

Additional simulations have been performed to report the effect of not including relaxation in the calculations of the interaction energy, summarized in the supplementary information and Table S2:

“The lateral expansion of the composite supercell was 0.13%, and the interaction energy increased to 0.135 eV when relaxation was not allowed, i.e., all cell parameters fixed to α -MnO₂. The C₆H₂ showed a smaller difference in interaction energy between fixed and relaxed lattice parameters, in line with the minor lateral expansion of 0.02% (Table S2).”

Table S2: Interaction energy and structural changes upon encapsulation in α -MnO₂. The lattice parameters were either fixed to α -MnO₂ or relaxed as given by the lateral expansion. The strain refers to the c-axis of α -MnO₂. Values are given for the AFM2 configuration.

Material	Diameter (Å)	Lateral expansion (%)	Strain (%)		Interaction energy (eV)
			α -MnO ₂	C ₉	
α -MnO ₂ + C ₆ H ₂	4.99	–			0.109
	5.04	0.46			0.090
α -MnO ₂ + C ₉	4.97	–	–	–1.41	0.135
	5.08	0.92	0.13	–1.28	0.065

It may be noted that the channel diameter can expand upon encapsulation of carbon chains even with fixed lattice parameters since the chains are only introduced in half of the channels which allows some relaxation of the local structure of the channel.

4. How does the axial position of C₉ affect the energy of the composite system? Does it exhibit energy barriers for displacement similar to the ones observed in C₆H₂, as shown in Figure 3?

Author reply: C₉ exhibits a very minor barrier and this was briefly mentioned in the main manuscript (P4, after Fig.2): “The calculated energy barrier for displacement of cumulenic C₉ along the c-axis of α -MnO₂ was miniscule (1.2 meV).”

The figure showing the energy profile has now been included in supplementary information:

“Figure S1 shows the energy barrier for displacement of cumulene along the α -MnO₂ channel. The obtained barrier of 1.2 meV is miniscule compared to the barrier of about 1 eV for C₆H₂ (see Figure 3).”

Figure S1: **Energy barrier for displacement of cumulenenic C₉ along the c-axis of α -MnO₂ from CI-NEB.** The dashed line is a guide to the eye. The data were obtained with the AFM2 configuration.

The local minima in Figure S1 has not been discussed further due to the very small energy differences compared to the C₆H₂ barrier.

5. Table S2 indicates relatively large changes in vibrational frequencies along the c-axis, suggesting that encapsulation affects the zero-point vibrations of oxygen anions. Did the author also calculate the zero-point vibrations of the C₉ chain, and if so, how does it impact Raman activity?

Author reply: Calculations of the vibrational frequencies of the C₉ chain upon encapsulation have been pursued, but not been completely successful. There are some changes in the frequencies, e.g., a lateral vibrational mode was 383.2 cm⁻¹ when encapsulated and 386.1 cm⁻¹ for the fully relaxed free-standing chain. However, the strain induced upon encapsulation has a significant effect on vibrational frequencies, and accurate simulation of the effect of encapsulation are difficult to differentiate from effects due to the artificial strain. Thus, it was concluded that the current computational setup is not suited for describing changes in the vibrational modes of C₉ upon encapsulation.

6. In addition to first-principles calculations, MD results have also confirmed that carbon chains can be inserted into CNTs with diameters favorable for vdW interactions. For example, relevant publications such as "Nanoscale, 2023, 15, 6143 (10.1039/D3NR00386H)" and "Carbon 2021, 183, 571 (10.1016/j.carbon.2021.07.037)" provide further support for this observation and could be referenced in the introduction section.

Author reply: Thank you for these suggestions. Among these references, the second was found most relevant to the context of the manuscript, and this was added as ref. 16 in the main manuscript.

Reviewers' comments:

Reviewer #1 (Remarks to the Author):

The author satisfactory replied to all of my comments and modified the manuscript accordingly. So, I can recommend publication of this manuscript in Communications Chemistry.

Reviewer #2 (Remarks to the Author):

The interaction energies were used for comparisons in Table 1 and Table S2, however, the interaction energy defined in this work depends on the length of carbon chain (or the number of C atoms). So the normalized interaction energy per C atom should be used for comparison (H effect should be excluded). Otherwise, many conclusions according to the comparisons may be misleading.

Response to the Reviewers

The author is grateful for the further review of the manuscript. The reviewers' comments and point-by-point responses are included below together with the related changes made to the manuscript and supplementary information.

Reviewer #1

The author satisfactory replied to all of my comments and modified the manuscript accordingly. So, I can recommend publication of this manuscript in Communications Chemistry.

Reviewer #2

The interaction energies were used for comparisons in Table 1 and Table S2, however, the interaction energy defined in this work depends on the length of carbon chain (or the number of C atoms). So the normalized interaction energy per C atom should be used for comparison (H effect should be excluded). Otherwise, many conclusions according to the comparisons may be misleading.

Author reply: In the original submission, both the total interaction energy and the interaction energy per carbon atom was included. However, in the revised manuscript, only the interaction energies per carbon atom were used in order to avoid confusion and facilitate comparison between systems. The text consistently refers to interaction energies per carbon atom. This has been clarified in the methods section:

“Interaction energies were calculated as the total energy difference between the composite supercell and the isolated materials, and normalized per carbon atom.”

This is also commented in the caption of Table 1 and now added to the caption of Table S2:

“The interaction energies are given per carbon atom”